# Dysprosium Absorption of Aluminum Tolerant- and Absorbing-Yeast

**Masao Kishida [1,2,*] and Kosuke Kakita [1]**

[1] Department of Applied Biological Chemistry, Graduate School of Agriculture, Osaka Metropolitan University, Gakuen-cho 1-1, Naka-ku, Sakai 599-8531, Osaka, Japan; hiro_ks2000@yahoo.co.jp

[2] Research Center of Microorganism Control, Osaka Metropolitan University, Gakuen-cho 1-1, Naka-ku, Sakai 599-8531, Osaka, Japan

[*] Correspondence: masyksd@omu.ac.jp; Tel.: +81-72-254-9455

**Abstract:** Background: Biosorption plays important roles in the recovery of rare earth metals. The absorption of dysprosium (Dy) was tested in yeast. Interestingly, brewing yeast, *Saccharomyces cerevisiae*, showed Dy absorption, and two strains, Alt-OF2 and Alt-OF5—previously isolated as highly aluminum-tolerant and -absorbing yeast strains—were screened and shown to be superior in terms of their Dy absorption when compared to *S. cerevisiae*. Here, we analyzed the Dy absorption in these yeast strains. Methods: Dy absorption in yeast strains was measured using an inductively coupled plasma optical emission spectrometer (ICP-OES). Dy concentration and localization in yeast cells and the effect of treated pH on the Dy absorption were assayed. Results: The Dy absorption of Alt-OF2 and Alt-OF5 was more than two times that of *S. cerevisiae*. The absorption of Dy took place inside of the cells, and a small amount was found in the cell wall fraction. Conclusion: These results suggest that yeast offers a promising solution to the biosorption of rare earth metals and that it is possible to use the highly absorbent strains to breed a yeast strain that can recover even higher concentrations of Dy.

**Keywords:** biosorption; dysprosium; yeast; screening; metal accumulation

## 1. Introduction

Rare metals, including lithium, beryllium, titan, vanadium, chromium, manganese, cobalt, nickel, gallium, germanium, rubidium, strontium, zirconium, niobium, molybdenum, palladium, indium, antimony, tellurium, cesium, barium, hafnium, tantalum, tungsten, rhenium, platina, thallium, bismuth, and rare earth elements (REEs) composed of scandium, yttrium and 15 lanthanide elements, are globally scarce and are difficult to replace with other metals in many cases [1]. In particular, REEs are due to the useful features they possess, rare metals are also important for high-tech industries, such as the automobile industry, digital consumer electronics, and information-related devices because they exhibit magnetism, fluorescence, and superconductivity due to their characteristic electron orbitals [2,3]. As powerful magnetic and superconductive materials are essential for the development of advanced industries that require high energy efficiency, REEs are indispensable ingredients in the manufacture of these products. For these reasons, rare metals, including REEs, are called the vitamins of industry, and the demand for rare metals is increasing due to the ongoing development of these high-tech industries [1]. Recently, the international prices of rare metals have markedly increased because they are required by many members of the high-tech industry, but their supply system is weak. A stable supply system for rare metals is crucial for the development of global science and technology industries. Moreover, large amounts of rare metals are accumulated in high-tech products, and they can be regarded as resources. Therefore, the efficient recovery of rare metals from nature, wastewater, and discarded high-tech products is very

important in terms of resource recovery, environmental cleanup, and also sustainable development goals (SDGs). Recently, the biosorption of REEs from REE-containing materials has become a booming process due to its high efficiency, cost effectiveness, versatility, and non-involvement of hazardous chemicals compared to physical or chemical processes, such as co-precipitation, solvent extraction, solid-phase extraction, and ion exchange, which include different organic–inorganic pollutants, toxic metals, and metalloids [4,5]. Biosorption uses biological methods, such as the metal-binding capacities of different biological materials present in living and non-living biomass, e.g., plant materials, industrial, and agricultural waste, as well as microorganisms such as algae, fungi, bacteria, and yeasts, and recent literature reports have focused on biosorption for the removal of rare earth metals present in bacterial culture [6], the batch culture of brown algae [7] and the biomass of plants [8]. However, the more promising works seem to explore the removal and recovery of REEs from very dilute solutions by biosorption.

Lanthanides are highly limited resources, and there is a need to establish highly technological apparatus for their removal [9]. Easy and ecological recovery systems for lanthanides would be incredibly useful for the building of smart cities. One particular REE, dysprosium (Dy), is experiencing a rise in demand due to its heat-resistant and powerful magnetic properties. In this study, we attempt to isolate yeast strains that can absorb Dy in order to establish a new recovery system for Dy using the bioresources of liquid waste with low Dy concentrations, such as mine drainage, ore waste, and industrial waste solutions. In this paper, we describe the screening of Dy-absorbing yeast strains and the characteristics for Dy absorption in the selected aluminum (Al)-tolerant and -absorbing strains, which have been previously isolated [10].

## 2. Materials and Methods

### 2.1. Strains, Media and Culture Conditions

The yeast strains used were *Saccharomyces cerevisiae* BY4741 [11], which were provided by Open Biosystems (Lafayette, CO, USA), as the standard for the popular yeast and *Schizoblastosporion* sp. Alt-OF2 and Alt-OF5, isolated as strains that are tolerant to and capable of absorbing high concentrations of Al, from a vegetable farm near the city of Takaraduka in Japan [10]. The culture medium used was GYP (2.0% glucose, 1.0% peptone, and 0.5% yeast extract at pH 5.0). When solid media were required, 2.0% of agar was added. To test the effect of Dy on yeast growth, Dy-acetate was added at 0.1, 1.0, and 10.0 mM in GYP. Culture conditions were kept at 30 °C.

### 2.2. Preparation of Yeast Protoplasts

To prepare yeast protoplasts, the yeast cell wall was solubilized using a β-1, 3-glucanase mixture, Zymolyase 100T (Nacalai Tesque Inc., Tokyo, Japan). The treatment condition was in accordance with the method described by Kishida et al. [12] and Li and Karboune [13], with some modifications, as follows. Yeast cells were treated with 1 mg of enzyme (approximately 1000 units) per $1.0 \times 10^7$ cells at 37 °C for 1 h in 50 mM potassium phosphate buffer (pH 7.0), and 1 M sorbitol was added. After the enzyme treatment, yeast protoplasts (precipitates) and the solubilized cell wall fraction (supernatants) were fractionated using centrifugation at approximately $2000 \times g$ for 5 min. Yeast protoplasts were collected by centrifugation at approximately $2000 \times g$ for 5 min after washing once using ultrapure water with 1 M sorbitol.

### 2.3. Preparation of Dead Cells

Yeast cells, cultured in GYP at 30 °C for 24 h, were collected by centrifugation with $5000 \times g$ for 10 min, washed twice with ultrapure water, and suspended with ultrapure water. The yeast suspension was treated with heat at 70 °C for 1 h. After the heat treatment, yeast cells were recollected as dead cells. Yeast cell killing ensured that the colonies were not recognized on the GYP plate for 7 days culture after spreading onto the plate.

*2.4. Assay for Dy Absorption in Yeast Cells*

Yeast strains were cultured in GYP at 30 °C for 24 h with shaking, and then yeast cells were collected by centrifugation with $5000 \times g$ for 10 min. After washing twice with ultrapure water, yeast cells reached an optical density of 600 nm, with a wavelength light ($OD_{600}$) equal to 10.0, and were suspended with 50 mL of 0.01 M sodium acetate buffer containing 0.1 g/L (equal to $6.09 \times 10^{-4}$ M) of Dy acetate. Buffer pHs were changed from 3.0 to 7.0 under different assay conditions. The Dy absorption of yeast cells was carried out with mild stirring at 30 °C for 1 h. After Dy treatment, whole cells were collected by centrifugation with $5000 \times g$ for 10 min, washed twice with ultrapure water, and suspended with ultrapure water. Yeast cells reached an $OD_{600}$ equal to 1.0, and were recollected and digested using 6 M pure nitric acid for approximately 1 h at 90 °C. If Dy-treated yeast cells were prepared as protoplasts, 1 M ultrapure water-solubilized sorbitol was used for washing and suspension before nitric acid digestion. After 50–100-fold dilution with ultrapure water, the Dy concentration of the samples was measured using an inductively coupled plasma optical emission spectrometer (ICP-OES), VISTA-MPX (Hitachi High-Technologies, Tokyo, Japan). All samples were analyzed at least in triplicate, and data were reported as mean values. Yeast cell numbers were calculated using the reported value of approximately $1.5 \times 10^7$ cells/mL suspension for the yeast cell concentration of 1.0 $OD_{600}$ [14].

**3. Results**

*3.1. Effect of Dy on Yeast Growth*

The effect of Dy on yeast growth was assayed. Growth curves were tested under various concentrations of Dy (Figure 1). The typical yeast, *S. cerevisiae*, showed more repressive growth under a higher concentration of Dy (Figure 1A), whereas the repressive growth was not recognized in two Al-tolerant strains, Alt-OF2 and Alt-OF5, under the Dy concentration that reached 10 mM (Figure 1B,C, respectively). These results suggest that Dy causes the inhibition of growth of the typical yeast, but the Al-tolerant strains, Alt-OF2 and Alt-OF5, have some mechanisms to allow their growth in a manner that is tolerant to the injury caused by Dy. This suggests that both Al-tolerant strains have a higher Dy absorbing ability, as the higher Al absorption of their strains is predicted to be related to their Al-tolerant ability.

*3.2. Dy Absorption in Yeast Strains*

Two Al-tolerant strains, Alt-OF2 and Alt-OF5, were tested for Dy absorption. The brewing yeast *S. cerevisiae* was used as the standard yeast for Dy absorption, and the change profiles for the Dy contents in yeast cells are shown in Figure 2. Alt-OF2 and Alt-OF5, which showed higher Al absorption, rapidly absorbed Dy for approximately two hours and remained at the stationary level for up to four hours; after this, they exhibited stationary absorption (approximately 2.0 pg/cells). On the other hand, interestingly, the *S. cerevisiae* strain, BY4741, interestingly appeared to be able to absorb Dy less than Alt-OF2 and Alt-OF5; however, the Dy absorption in *S. cerevisiae* is rarely reported. These results suggest that yeast strains have the ability to absorb Dy and that Al-absorbing strains also have a greater ability to absorb Dy than the popular yeast *S. cerevisiae*. Moreover, these results also suggest that the mechanisms involved in Al tolerance or absorption have an effect on the Dy absorption because Alt-OF2 and Alt-OF5, isolated as Al-tolerant and -absorbing yeast strains, exhibit more Dy absorption than BY4741.

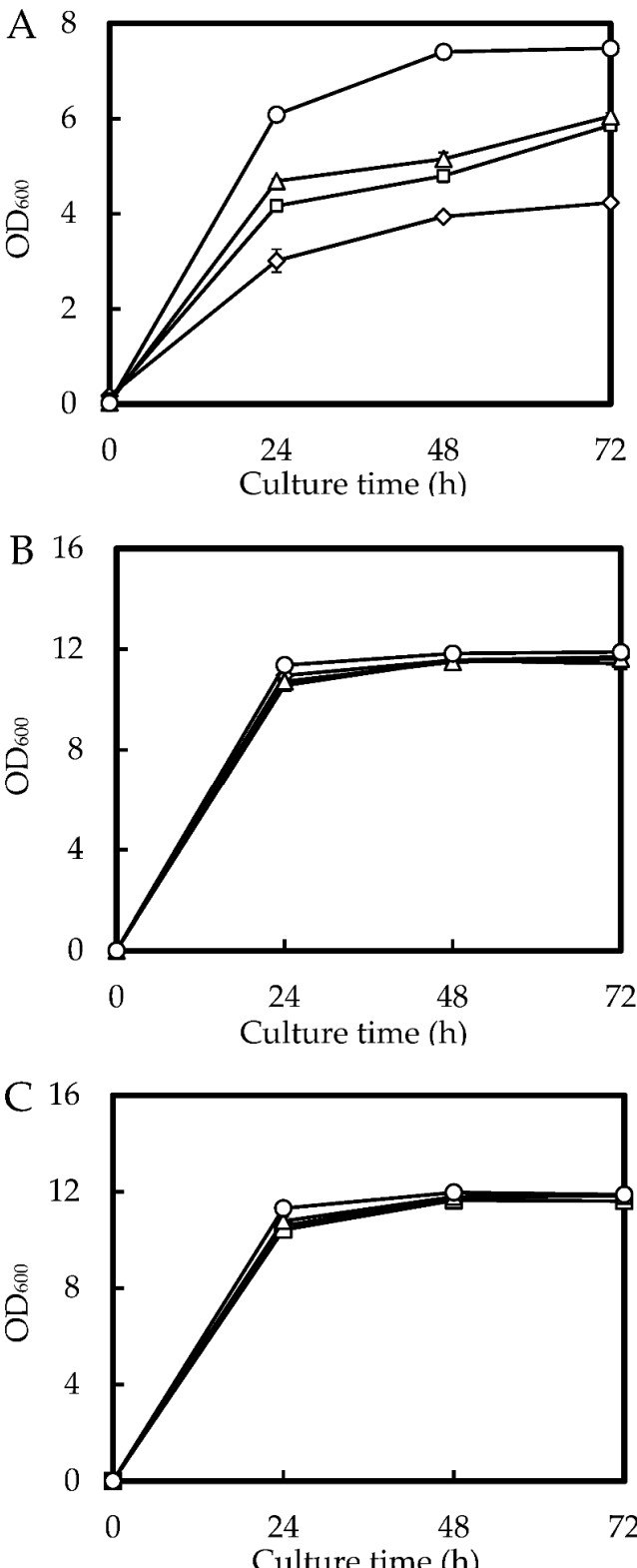

**Figure 1.** Growth profiles of yeast strains in Dy-containing media. Yeast strains were cultured in medium containing Dy at 0 mM (circles), 0.1 mM (triangles), 1.0 mM (squares), and 10 mM (diamonds) at 30 °C with shaking. (**A**) *S. cerevisiae* BY4741; (**B**) Alt-OF2; (**C**) Alt-OF5.

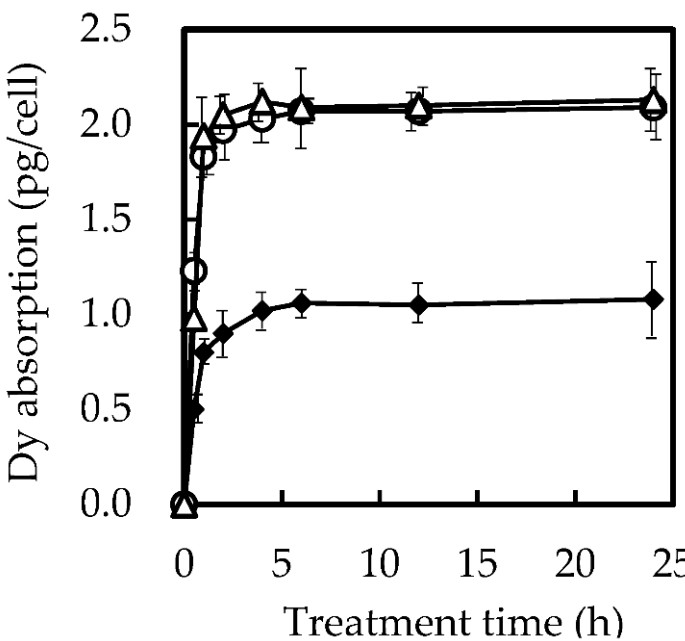

**Figure 2.** Profiles of Dy absorption based on exposure time. Filled diamonds are *S. cerevisiae* BY4741. Open circles and triangles are Al-absorbing strains Alt-OF2 and Alt-OF5, respectively.

*3.3. Effect of Treated pH on Dy Absorption*

Testing the Dy absorption of Al-tolerant strains at pH 3.0 to 7.0, it was shown that Alt-OF2 and Alt-OF5 were most efficiently absorbed dysprosium under pH 6.0 (Figure 3). At the lower pH (3.0–5.0), Dy absorption was lower. These results suggest that any activated factors at approximately pH 6.0 were shown in the yeast cells, whereas Dy absorption at the lower pH tended to increase when yeast cells were treated for a longer time at that pH (data not shown). These results suggest that the factors involved in Dy absorption do not affect its activity.

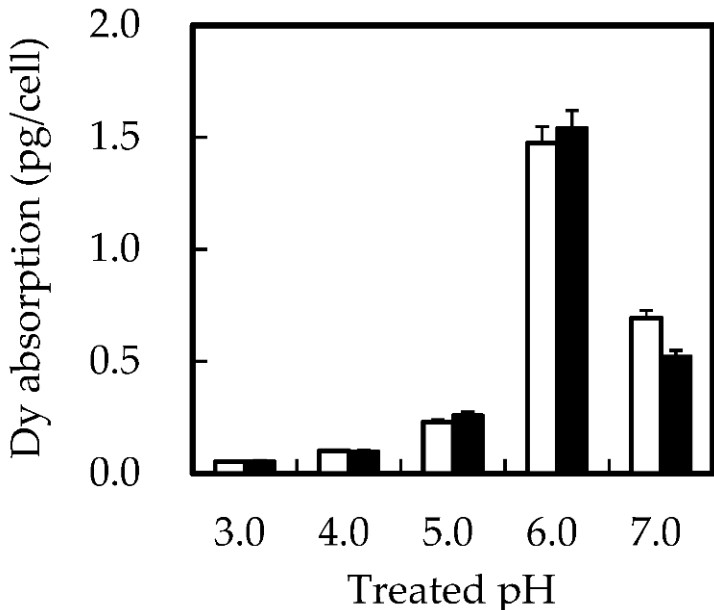

**Figure 3.** Effect of treated pH on Dy absorption in Alt-OF2 and Alt-OF5. Yeast (Alt-OF2 and Alt-OF5) cells were treated with 10 mM sodium acetate buffer at pH 3.0–7.0 containing 0.1 g/L of Dy acetate for 2 h. White and black columns represent Alt-OF2 and Alt-OF5, respectively.

### 3.4. Dy Absorption Using Heat-Killed Cells

When the dead yeast cells treated with heat at 60 °C were tested for Dy absorption, the amount of Dy in the dead cells was approximately 50% of that in the untreated intact cells of both yeasts (Figure 4). These results suggest that some factors involved in Dy absorption are bioactive materials, and the others result from physicochemical events, such as cell surface adsorption.

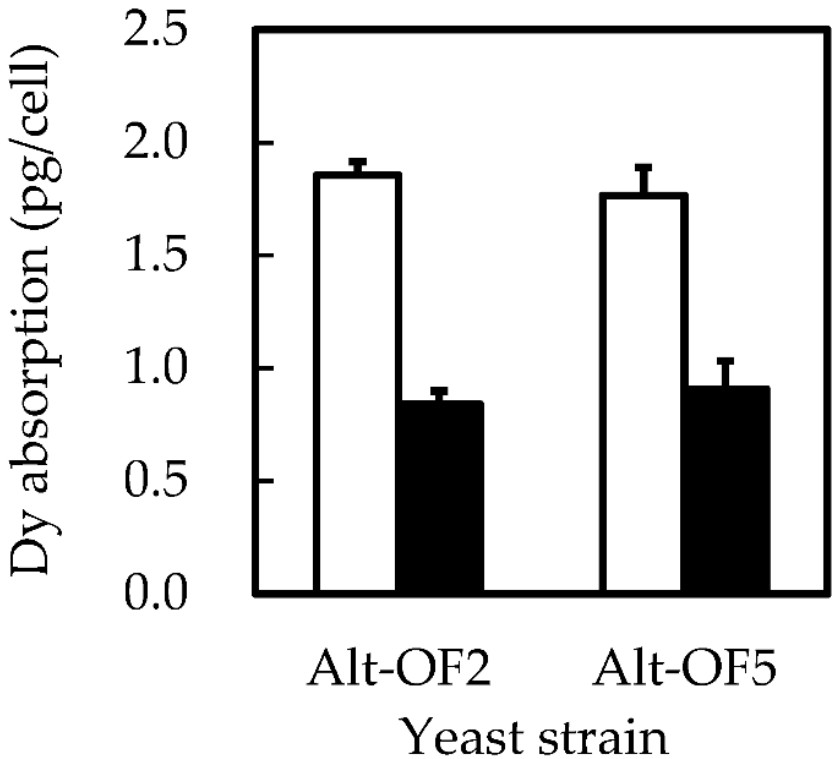

**Figure 4.** Dy absorption in the heat-killed cells of Alt-OF2 and Alt-OF5. Yeast cells (Alt-OF2 and Alt-OF5) were treated at 60 °C and then treated with 0.1 g/L of Dy acetate at pH 6.0 for 2 h. White and black columns represent untreated and heat-treated cells, respectively.

### 3.5. Localization of Absorbed Dy

To elucidate whether absorbed Dy is localized on the surface of the cell wall or inside of the cell, the protoplast fraction and the solubilized cell wall fraction were separately prepared from the Dy-absorbing Alt-OF2 by treatment with a cell wall-solubilizing enzyme, Zymolyase. The Dy content was low (less than 0.2 pg per yeast cell) in the solubilized cell wall fraction, but that in the protoplast fractions showed a similar profile to that in the intact cells (Figure 5). These results suggest that the majority of Dy is taken up into the inside of the cells in yeast, especially in the Al-tolerant and -absorbing strains. These results differ from those found regarding the metal absorption of bacteria.

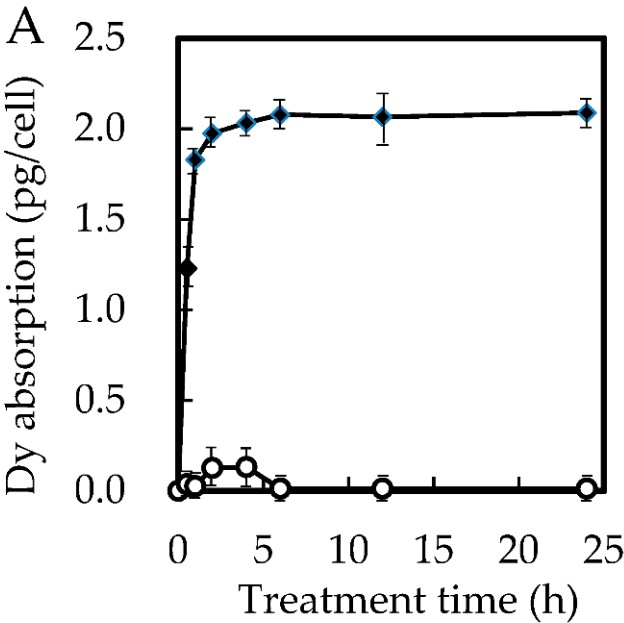

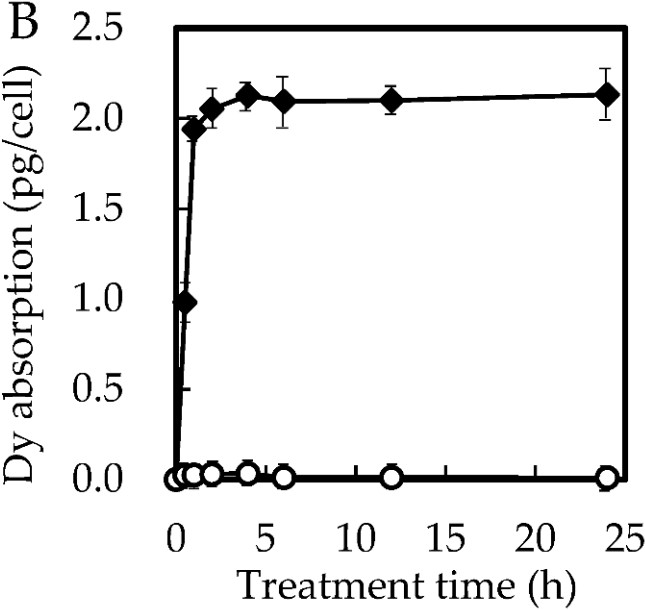

**Figure 5.** Dy content of protoplasts and solubilized cell wall fraction in the Alt-OF2 (**A**) and Alt-OF5 (**B**). Yeast cells were treated with 0.1 g/L of Dy acetate for 1 h, collected, and then digested with Zymolyase 100-T for 1 h. Closed diamonds and open circles represent the Dy concentration in protoplasts and the solubilizing cell wall fraction, respectively.

## 4. Discussion

In this study, we successfully selected three Dy-absorbing yeast strains, one being the *S. cerevisiae* laboratory strain, BY4741, and the other two the highly Al-tolerant and -absorbing strains Alt-OF2 and Alt-OF5. These may be used to recover Dy from the waste solutions produced following the metal manufacturing process. The concentration of REE absorption is from 0.5 to 1.0 mmol per 1 mg biomass, as reported in biosorption using various biomaterials [6–9], and in recent research on the biosorption of REE using recombinant bacteria [15] and moss [16]. The recovery rate of REE absorption is reported to be 80–90% [6,9,15]. Our result shows that the Dy absorption of yeast is 1.0–2.0 pg per cell (in

Figure 2). When the yeast biomass was measured in our strains, we found 0.65–0.75 mg dry weight per 1 $OD_{600}$ unit ($OD_{600}$ = 1), similar to the reported value [17]. The calculated Dy concentration absorbed in 1 mg (dry weight) of our yeast biomass was 0.2–0.4 mmol. These values are by no means inferior to those when using other biomaterials, considering that the treatment in the Dy solution lasted 2 h. The recovery rate of Dy using our Al-tolerant and -absorbing yeast strains was more than 90%, similar to that using recombinant bacteria. Our yeast strains are more easily handled than bacteria containing recombinants in terms of biosafety. In particular, *S. cerevisiae* is especially useful, if possible, because it is regarded as safe to use in the processing of food and beverages.

It is strongly suggested that Dy absorption using the Al-tolerant and -absorbing yeast strains can establish a new recovery system. However, the mechanism of Dy absorption remains unknown in yeast. Our research results suggest that Dy is adsorbed in the cell surface region because Dy absorption was recognized using dead yeast cells. However, the Dy concentration using the dead yeast cells was 50% of that of the intact yeast cells. Our preliminary experiments suggest that Dy absorption is decreased to a similar level using dead cells, when the Dy treatment is carried out at a high temperature at which the yeast strains cannot grow (our unpublished data). These results suggest that bioactive factors are involved in Dy absorption. Additionally, this suggestion is supported by our results that the Dy absorption is decreased at a lower pH (in Figure 3), at which the biological activity is lower. The Dy concentration of the Dy-absorbing yeast cells when removing the cell wall was similar to that of the intact Dy-absorbing cells. These results may also show that Dy that is present in the cell surface region is located on the cell membrane and is transported into the intracellular region by bioactive factors.

The Dy-absorption mechanisms of yeast have not yet been determined. However, the mechanisms involved in Al tolerance or absorption clearly contribute to the Dy absorption in yeast because the Al-tolerant and -absorbing strains, Alt-OF2 and Alt-OF5, show higher Dy absorption. Our results also support the belief that ionized Dy produces the same trivalent ion as Al. If the mechanism of Dy absorption in yeast is elucidated, our results may contribute not only to the study of the mechanism underlying the biosorption of Dy but also to elucidating the unknown mechanisms of the yeast response to rare earth metals because *S. cerevisiae* is an ideal model organism [18]. In terms of the yeast response to Al, it is interesting to note that the deletion mutants of the Al transport-related genes that display an Al-tolerant phenotype harbor an Al transporter gene owing to the low probability of Al accumulation in the cell wall. Moreover, Al-absorbing strains do not die upon intracellular Al accumulation, and Al cytotoxicity is not observed in *S. cerevisiae* strains with deletions in Al transport-related genes [19]. Further studies may be carried out based on our results on the Dy absorption of *S. cerevisiae*. Although it remains unknown as to whether transporter-related proteins can act on trivalent ions such as Al, future studies should focus on identifying the transporter responsible for importing these trivalent ions into cells.

## 5. Conclusions

The Al-tolerant and -absorbing yeast strains, Alt-OF2 and Alt-OF5, are capable of the biosorption of rare earth metals such as Dy. Further physiological and molecular genetic studies on Dy uptake into yeast cells are recommended in order to overcome the present barriers to their application; nonetheless, we expect that these strains will be applicable for the biorecycling of valuable rare earth metals such as Dy and thus contribute to the SDGs.

**Author Contributions:** Conceptualization, M.K.; resources, M.K.; data curation, K.K.; writing—original draft preparation, K.K. and M.K.; writing—review and editing, M.K.; supervision and development, M.K. All authors discussed the experimental results and methodology and contributed to the preparation of the final draft. All authors have read and agreed to the published version of the manuscript.

**Funding:** This research received no external funding.

**Institutional Review Board Statement:** Not applicable.

**Informed Consent Statement:** Not applicable.

**Data Availability Statement:** Not applicable.

**Acknowledgments:** We kindly thank Masakazu Furuta (Research Center of Microorganism Control, Osaka Prefecture University) and Naoki Matsumoto (Environment Bureau, The City of Osaka) for their support with the mechanical analysis in this study.

**Conflicts of Interest:** The authors declare no conflict of interest.

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
