# Peer review of "Dysprosium Absorption of Aluminum Tolerant- and Absorbing-Yeast"

_applsci, doi:10.3390/app12094352_

Round 1
Reviewer 1 Report
The manuscript presented is interesting from the point of view of waste utilization. The biosorption of heavy and rare earth metals by the yeast is of great attention.
I suggest the publication of this work, but the major revision is required.
- In theexperimental section, the OD is described to be measured at 660 nm, but in Fig.1 the OD is signed as OD600. The authors should correct the wave length either in Section 2-4 or inFig. 1
- The data on the temperature influence on biosorption and yeast growth should be presented.
- The discussion of the biosorption mechanisms as well as the full discussion of the results should be given in the manuscript.
- The authors should clearly formulate the aim of the study. Also the gaps in the biosorption studies must be stated in the introduction of the manuscript. I suggest the authors to review the state of the arts works and refer the recently published papers.
- Also, the comparison of the results obtained with the literature data and other works (i.e. metals, microorganisms used etc.) should be presented while the discussing the results.
Author Response
- In theexperimental section, the OD is described to be measured at 660 nm, but in Fig.1 the OD is signed as OD600. The authors should correct the wave length either in Section 2-4 or inFig. 1
Sorry, it is our mistake. All data were measured at OD600. We correct OD660 (660 nm) to OD600 (600) in “2-4. Assay for Dy absorption in yeast cells” part..
- The data on the temperature influence on biosorption and yeast growth should be presented.
Our preliminary study shows that the Dy absorption in treatment at the temperature at which yeast cannot grow was similar to that using the dead cells. So This discussion “ Our preliminary studies…….yeast strain cannot grow (our unpubished data).” (p8, line 24, - line 26) is added.
- The discussion of the biosorption mechanisms as well as the full discussion of the results should be given in the manuscript.
We revise introduction.
- The authors should clearly formulate the aim of the study. Also the gaps in the biosorption studies must be stated in the introduction of the manuscript. I suggest the authors to review the state of the arts works and refer the recently published papers.
- Also, the comparison of the results obtained with the literature data and other works (i.e. metals, microorganisms used etc.) should be presented while the discussing the results.
To resolve Q4-Q5, the discussion is increased (p8, line 10 - line 33).
Reviewer 2 Report
1. Introduction.
Should be re-written according to the main goal of the manuscript.
Boron, Selenium and Antimony are not metals!
2. Materials and Method.
Method for determination of cell numbers!
Figure 1. What kind of units is OD 660.
Quantity of Dy absorbed per g of dry biomass?
Author Response
- Introduction.
Should be re-written according to the main goal of the manuscript.
The discussion is increased (from p8, line 10 to p8, line 33), according to the main goal of manuscript.
Boron, Selenium and Antimony are not metals!
These are excluded.
- Materials and Method.
Method for determination of cell numbers!
Figure 1. What kind of units is OD 660.
Sorry, it is our mistake. All data were measured at OD600. We correct OD660 (660 nm) to OD600 (600) in “2-4. Assay for Dy absorption in yeast cells” part..
Quantity of Dy absorbed per g of dry biomass?
Dy absorption per g is corrected Dy absorption per cells
Reviewer 3 Report
The authors evaluated the biosorption potential via using brewing yeast strains and found some interesting results. The experiment was well designed and conducted. The discussion part was too short, The authors may add more discussion by comparing your results with previous studies, comparing the differences, and explaining why?
Author Response
The authors evaluated the biosorption potential via using brewing yeast strains and found some interesting results. The experiment was well designed and conducted. The discussion part was too short, The authors may add more discussion by comparing your results with previous studies, comparing the differences, and explaining why?
The novel discussion is added in the discussion part (p8, line 10 - line 33). The comparison from our results and some previous studies is explained in this discussion,
Round 2
Reviewer 1 Report
I recommend to publish the revised manuscript.
Reviewer 2 Report
No.
Reviewer 3 Report
no further comment.